# Influence of Phosphate on Arsenic Adsorption Behavior of Si-Fe-Mg Mixed Hydrous Oxide

**DOI:** 10.3390/toxics12040280

**Published:** 2024-04-11

**Authors:** Marjjuk Ahmed, Tomoyuki Kuwabara

**Affiliations:** 1Graduate School of Natural Science and Technology, Shimane University, 1060 Nishikawatsu, Matsue 690-0823, Japan; 2Institute of Environmental Systems Science, Shimane University, 1060 Nishikawatsu, Matsue 690-0823, Japan; kuwabara@life.shimane-u.ac.jp

**Keywords:** arsenate, arsenite, anion selectivity, adsorption capacity, adsorption isotherm

## Abstract

The arsenic adsorption performance of silicon (Si), iron (Fe), and magnesium (Mg) mixed hydrous oxide containing a Si: Fe: Mg metal composition ratio of 0.05:0.9:0.05 (SFM05905) was investigated. SFM05905 was synthesized by the co-precipitation method. Batch experiments on arsenic adsorption were conducted at various temperatures and concentrations. Adsorption isotherms models were represented by a linearized equations and were insensitive to temperature change. The anion selectivity of SFM05905 at single component was high for arsenite (III), arsenate (V), and phosphate (PO_4_), indicating that PO_4_ inhibits arsenic adsorption. The adsorption amount of As (III), As (V), and PO_4_ were compared using a column packed with granular SFM05905, and an aqueous solution was passed by a combination of several anions that are single, binary, and ternary adsorbate systems. As (III) had the highest adsorption amount; however, As (III) and PO_4_ were affected by each other under the ternary mixing condition. Although the adsorption amount of As (V) was smaller than that of As (III), it was not affected by other adsorbates in the column experiments. Finally, although the adsorption of both arsenic continued, the adsorbed PO_4_ gradually desorbed.

## 1. Introduction

Arsenic, a highly toxic element, poses a significant threat to global health [1]. Its inorganic forms, arsenite (As-III) and arsenate (As-V), particularly in contaminated drinking water, present serious environmental and human health concerns [2]. The naturally occurring arsenic in drinking water allows its easy entrance into the body via groundwater. Chronic exposure to arsenic-contaminated water has been linked to various health issues, including skin lesions, respiratory problems, and neurological disorders [3,4,5]. Arsenic exists naturally in various oxidation states (−3, 0, +3, and +5) with arsenate and arsenite being the most common forms found in different environmental conditions [6]. Inorganic species of arsenic are commonly linked covalently with oxygen as arsenite (AsO_3_^3−^) and arsenate (AsO_4_^3−^) [7]. Arsenic contamination in groundwater has become a major issue in many countries, including Bangladesh, Vietnam, and the United States, necessitating stringent regulations for arsenic concentration in drinking water [8]. The toxicity of arsenic varies depending on its chemical form, with arsenite being the As (III) species is the most toxic, although arsenate is the most mobile [9]. Inorganic arsenic is more toxic than organic arsenic [10]. The main sources of arsenic include release from arsenic-bearing sediments in groundwater aquifers and human activities such as agricultural applications, mining, smelting, and other industrial activities. The World Health Organization (WHO) has revised the permissible limit for arsenic concentration in drinking water from 50 μg/L to 10 μg/L [11], highlighting the urgent need for efficient methods for arsenic removal from drinking water [12].

Various technologies are being employed for arsenic removal, including adsorption, coagulation, and membrane processes [13,14,15,16]. Among these, adsorption technology offers advantages such as simplicity, cost-effectiveness, and ease of regeneration [17].

The mixed hydrated oxides of silicon (Si), iron (Fe) and magnesium (Mg), which combine several types of inexpensive polyvalent metal elements, synthesized by the neutralization–precipitation method, have shown promise as adsorbents [18]. The arsenic adsorption performance of the mixed hydrous oxides with a Si, Fe, and Mg molar ratio of 1:8:1 and 1:1:8 (referred to here as SFM181 and SFM118, respectively) was published in a Japanese journal [19]. In the previous study, the adsorption ability of some SFM samples with different Fe/Mg ratios was investigated in order to adsorb low-concentration As (III) and As (V) in groundwater [20]. SFM118 and SFM181 had high adsorption amounts for As (V) and As (III), respectively. In the SFM series, by increasing the molar ratio of Fe content, the adsorption amount of As (V) did not change significantly, and the adsorption amount of As (III) increased. Furthermore, SFM with a large Fe/Mg ratio tends not to be inhibited by other anions. Since it was expected that the molar ratio was affected to the arsenic adsorption ability, it was necessary to change the molar ratio for better adsorption process. As a result, I tried using SFM05905 to decrease the amount of Si and Mg and increase the Fe molar ratio. Therefore, it is expected that the arsenic adsorption capacity can be improved by further increasing the molar ratio of Fe. The influence of different sorption parameters, including adsorbent dose, solution pH, contact time, and initial arsenic concentration, was thoroughly studied to obtain the maximum adsorption, which occurred at pH 4.0, with equilibrium achieved in 35 min. Various isotherm models, including Langmuir, Freundlich, Temkin, Elovich, Dubinin–Radushkevich, and Flory–Huggins, were utilized to explain the adsorption phenomenon [21].

In this paper, we investigated the arsenic removal ability of SFM05905 with a molar ratio of Si, Fe, and Mg of 0.5:9:0.5. Our choice of this specific molar ratio was based on a thorough exploration of various ratios, including Si:Fe:Mg = 0.1:0.8:0.1 and 0.1:0.1:0.8, as presented in the introduction. Additionally, we considered the adsorption of arsenic for SFMs with Si:Fe:Mg ratios of 0:1:0, 0.1:0.4:0.5, and 0.1:0.6:0.3 [22]. Furthermore, based on the results of examining the arsenic adsorption capacity of various Si:Fe:Mg ratios (0:1:0, 0.05:0.9:0.05, 0.1:0.8:0.1, 0.1:0.9:0, 0.2:0.8:0, 0:0.9:0.1, and 0:0.8:0.2) (unpublished), we evaluated in detail the arsenic adsorption ability of SFM05905 with a molar ratio of 0.05:0.9:0.05 in this study. The adsorption isotherm, anion selectivity of powder SFM05905, and arsenic removal ability of granular SFM05905 were examined using column methods. Additionally, we focused on the effect of PO_4_ adsorption inhibition on arsenic adsorption, observing arsenic uptake during both batch and column adsorption processes.

This research aims to contribute to the development of efficient arsenic removal methods, particularly in regions facing groundwater contamination issues. By elucidating the arsenic adsorption behavior of SFM05905 and understanding the factors influencing its performance, this study seeks to provide insights for the design and optimization of arsenic removal processes in water treatment applications.

## 2. Materials and Method

### 2.1. Synthesis of Si-Fe-Mg Mixed Hydrous Oxide

Sodium silicate solution No.3 (Na_2_O•nSiO_2_ xH_2_O, Kishida Chemical, Osaka, Japan), FeCl_3_•6H_2_O, MgCl_2_•6H_2_O, and hydrochloric acid by Fujifilm Wako Pure Chemical Corp. (Osaka, Japan) were used for the synthesis process. Milli-Q purification system (Merck Millipore, Darmstadt, Germany) was used for the synthesis process and all water quality analyses.

The powder SFM05905 sample was synthesized by the neutralization–precipitation method which followed the method of an previous study [20]. The metal element composition ratio of SFM05905 was measured by the dissolution method. The gel formed after dissolution in nitric acid and perchloric acid while heating was collected with a quantitative filter paper (5B, Toyo Roshi Kaisha, Ltd., Tokyo, Japan), and a filtrate was obtained. After washing the filter paper, the dry weight of the gel was measured to determine the SiO_2_ content. The filtrate was analyzed by inductively coupled plasma mass spectrometry (ICP-MS) (Agilent8800, Agilent Technologies, Inc., Santa Clara, CA, USA) to measure Mg and Fe content.

### 2.2. Characterization of Powder SFM05905

The powder SFM05905 was characterized by X-ray diffractometer (Mini Flex II, Rigaku, Tokyo, Japan): excitation voltage of X-ray tube, 40 kV; current, 40 mA; Ni (β); Cu-Kα rays that passed through a filter were used; the scanning range was 2θ = 5 to 80°; scanning width, 0.02°; scanning speed, 2°/min.

### 2.3. Batch Adsorption for Adsorption Isotherm Using Powder SFM05905

The adsorption process was carried out in batches. A stock solution of arsenic (As) (ca. 1000 mg/L) was prepared with various concentrations (10, 20, 50, 100, and 150 mg/L) by dissolving dibasic sodium arsenate (Na_2_HAsO_4_.7H_2_O), which was obtained from Fujifilm Wako pure chemical corporation (Osaka, Japan), and Diarsenic tri oxide (As_2_O_3_), obtained from Kanto chemical Co., Inc. (Tokyo, Japan) and of analytical grade, in deionized water. In the experimental studies, working solutions were freshly prepared from the stock arsenic solution for each experimental run. The stock solution was further prepared with suitable concentrations, and the sample pH (6.8) was adjusted by using 1 mol/kg HCl solution and 1 mol/kg NaOH solution. All the batch adsorption experiments were carried out at various temperatures (25 °C, 35 °C, and 45 °C) by using Mixed Rotor apparatus (VMRC-5, AS ONE, Osaka, Japan). SFM05905 material was added in a concentration of 0.5 (*w*/*v*) % into the arsenic containing various solutions, and the resultant mixture was shaken for 72 h in three replicates. The sample solution was filtered using a membrane filter (HP020AN, Advantec, Tokyo, Japan) with a pore size of 0.2 μm, and the arsenic concentration was measured by inductively coupled plasma mass spectrometry (ICP-MS) (Agilent8800, Agilent Technologies, Inc., California, USA). The pH of the filtrate was measured by a pH meter (LAQUAtwin AS-712, HORIBA, Kyoto, Japan). The adsorption amount of arsenic ion adsorption per 1 g of SFM05905 sample was calculated by Equation (1), which is based on the difference between the initial concentration and the concentration after stirring:W_g_ = V (C_0_ − C_t_)/W (1)
W_g_: adsorption amount per 1 g of SFM05905 (mg/g); C_0_: initial concentration (mg/L); C_t_: concentration in sample water after t hours of shaking (mg/L); V: water volume of the sample (L); W: added quantity of SFM05905 (g).

Adsorption isotherms of As (III) and As (V) were constructed at each temperature. The effect of temperature on and equilibrium time and on the adsorption amount of As (V) and As (III) was investigated over the powder SFM05905 adsorbent.

### 2.4. Examination of Anion Selectivity for Powder SFM05905

The anion selectivity of each SFM05905 was examined using the batch method. The target ions were AsO_3_, AsO_4_, PO_4_, SO_4_, F, NO_3_, and CO_3_ ion, for a total of 7 types. The reagents shown in Table 1 were added and dissolved in purified water to adjust the concentration of each ion to 0.05 mmol/L, and then adjusted to pH 7.0 using 1 mol/L HCl and 1 mol/L NaOH. Each SFM05905 was added to each sample water so as to be 0.02 (*w*/*v*) %, and a constant speed circular stirring was performed at 190 rpm using a shaker (MMS-3020, Tokyo rikakikai, Japan) at room temperature. In the CO_3_ ion sample water, nitrogen gas was sealed in the container space immediately after the addition of SFM05905 to prevent carbon dioxide from dissolving in the sample water. The filtrate was collected through a 0.20 µm membrane filter before stirring and after arriving at equilibrium adsorption. The concentrations of As (III) ion and As (V) ion, as well as PO_4_ ion, were determined using ICP-MS spectrometry (Agilent 8800, Agilent Technologies, Inc., Santa, Clara, CA, USA). Fluoride ion, sulfate ion, and nitrate ion concentrations were measured using the ion chromatograph method (Ion chromatograph ICS-1600, Dionex, Sunnyvale, CA, USA), and carbonate ion concentrations were determined using the combustion/infrared analysis method with the TOC-V CSH/CSN analyzer (SHIMADZU, Kyoto, Japan). The adsorption amount of each ion adsorbed per 1 g of each SFM05905 was calculated by Equations (1) and (2). The adsorption amount of unit per volume was converted by Equation (3). In addition, the partition coefficient (K_d_) of each SFM05905 was calculated and evaluated by Equation (4). However, it is assumed that all the decrease in adsorbate is adsorbed by SFM05905.
W_m_ = W_g_/A_w_
(2)
C_r_ = W_mρ_
(3)
K_d_ = C_r_/C_s_
(4)
W_m_: adsorption amount per 1 g of SFM05905 (mmol/g); A_w_: atomic weight of adsorbate; ρ: true density of each SFM05905 sample (g/cm^3^); C_r_: solid phase ion concentration (mmol/cm^3^); C_s_: liquid phase ion concentration (mmol/cm^3^)

### 2.5. Preparation of Granular SFM05905

The granulation of SFM05905 was based on the method of carrying Iron Hydroxide Oxide on Polyacrylamide Cryogel reported by [22]. Monomeric acrylamide (AAm) and ammonium peroxydisulfate (APS) were obtained from Fujifilm Wako Pure Chemical Corp. (Osaka, Japan). N,N-methylene bis acrylamide (MBAAm) and N,N,N;N-Tetramethyl ethylene diamine (TEMED) were obtained from Alfa Aesar Ward Hill, MA, USA and Combi-Blocks, San Diego, CA, USA, respectively. SFM05905 powder and degassed water were used for the polyacrylamide granular SFM05905.

Firstly, 200 mL of pre-degassed water was put on a 500 mL beaker and then cooled down with ice until 0 °C. SFM05905 (20 g) was dispersed in degassed water while stirred by a magnetic bar. AAm (8 g) and MBAAm (2 g) were dissolved in the given order with vigorous stirring for 1.5 h, and then N_2_ gas was passed into the solution for 3 min. After that, APS solution, which comprises the APS (1.0 g) dissolved in 1.0 mL of degassed water, was added to the monomers solution, and stirred for 0.5 h. TEMED (0.24 g) was added quickly to the solution while stirring, and right after about 10 mL of the mixture solution was poured into some syringes. These syringes were sealed and put in a program-controlled refrigerator and frozen at −15 °C for 24 h. Finally, they were thawed in a warm bath and washed with distilled water until they reached neutral pH values. After that, the polyacrylamide (PAAm) containing SFM05905 was cut to an appropriate size (<1.0 cm^3^), which is shown in Figure 1, and dried with a freeze dryer (VD-800R, TAITEC, Koshigaya, Japan).

Lastly, the contents of SFM05905 in the granular form were determined using a heating and melting method. The sample was heated on a hot plate at 100–150 °C. If decomposition was not sufficient, a mixture of nitric acid and sulfuric acid was added. The mixture was then cooled, generating white smoke, and boiled slowly to settle insoluble matter. The resulting solution was filtered, and the Fe concentration in the filtrate of the melted sample was measured. Subsequently, the content of the powder SFM05905 in the granular form was calculated.

## 3. Column Adsorption of Arsenic with the Granular SFM05905

The continuous-flow arsenic adsorption experiment was conducted with the adsorption column method, which is shown in Figure 2; the experimental conditions of the column adsorption are shown in Table 2. A polyethylene column with a length of 20.0 cm and an inside diameter of ø 1.6 cm was used as an adsorption column. Dry granular SFM05905 was packed in the column with water to remove air from granular samples by applying negative pressure. Single [As (III)], [As (V)], and [PO_4_)], binary [As (III) + PO_4_], [As(V) + PO_4_], and [As (III) + As (V)], and ternary [As (III) + As (V) + PO_4_] adsorbate component systems of aqueous solution were adjusted to each anion concentration of 0.015 mmol/L and were passed in an upward flow from a silicon tube connected to the bottom of the column. Treated water was collected from the outlet at the top of the column at regular time intervals, and the residual arsenic concentrations and the pH were then measured. In this study, the space velocity (SV) was set to approximately 3.3/h.

## 4. Result and Discussion

### 4.1. Metal Element Composition Ratio XRD Analysis of SFM05905

The metal composition ratio of powder SFM05905 was 0.048 Si:0.907 Fe:0.046 Mg, which is shown in Table 3. The metal element composition ratio of powder SFM05905 was the same as that of the prepared solution for dissolving reagents during the synthesis of SFM05905. The carrying ratio of powder SFM05905 in the granular SFM05905 was calculated by comparing the Fe content. The Fe content of the granular SFM05905 in acid dissolution was 5.9 mmol/g, and that of the powder SFM05905 was 10.5 mmol/g. Thus, the Fe ratio of the granulate to the powder was 56.0. That is, the carrying ratio of SFM05905 in granular SFM05905 was calculated to be 56.0 wt.%.

In a previous paper, the authors showed the correlation coefficient relationship between the metal element composition ratio of each Si-Fe-Mg-based sample and that of the adsorption amount of the various ions. In addition, the correlation coefficients of the arsenite ion equilibrium adsorption amount with the Mg ratio and the Fe ratio, respectively, and the same positive correlation was observed for both the Mg ratio and the Fe ratio [23]. This indicates that arsenite ion adsorption is affected by both the Mg ratio and Fe ratio.

SFM05905 was analyzed by XRD to assess the physicochemical characteristics of the materials. The XRD patterns of the powder SFM05905 are shown in Figure 3. The XRD patterns of SFM05905 were not specific, and no diffraction peaks. In a previous report, based on the X-ray diffraction results, layered double hydroxide peaks were clearly detected in the SFM samples with a small Fe/Mg ratio such as SFM118, but when the Fe/Mg ratio increased, the specific peaks were could not be observed [20]. It was also shown that the main component of SFM010 synthesized using only Fe was α-FeOOH (Goethite). Therefore, the main component of SFM05905 was assumed to be amorphous Fe hydrous oxide.

### 4.2. FTIR Analysis of SFM05905

The FTIR spectrum shown in Figure 4 for SFM05905 composite materials reveal important features. At 3392 cm^−1^, a strong, wide band suggests the presence of hydroxyl (-OH) groups. Another significant signal at 1629 cm^−1^ indicates stretching vibrations in the C-H aromatic ring, confirming the presence of aromatic cycles in the extracts. Additionally, vibrations in the C–H region around 1100 cm^−1^ correspond to the C–O bond. The peak observed at 577 cm^−1^ is attributed to the Fe–O bond, further confirming the presence of SFM05905.

These peaks collectively confirm the presence of hydroxyl and Fe–O bonds, essential for the reduction of salt precursors to SFM05905 due to their lower redox potential. Specifically, the Fe–O bond at 577 cm^−1^ indicates the presence of SFM05905. These bonds likely originate from Fe-OH stretching and banding vibrations of hydroxyl groups, which are converted from iron oxide into forms such as Fe-OH, Fe(OH)_2_, or α-FeOOH (Goethite) phase. Importantly, these materials are known for their environmental friendliness [24].

## 5. Adsorption Isotherm

Adsorption isotherms for both As (III) and As (V) were obtained at several temperatures (25 °C, 35 °C, and 45 °C) and were shown in Figure 5. The measured adsorption amounts of As (III) and As (V) on SFM05905 were almost 25 mg/g at an equilibrium concentration of less than 2.5 mg/L and 25 mg/L, respectively. According to our previous study [23], the adsorption amount of As (III) on SFM118 was 10 mg/g at the equilibrium concentration of 1.3 mg/L. Similarly, the adsorption amount of As (V) was 15 mg/g at the equilibrium concentration of 24 mg/L. In other words, the As (III) and As (V) adsorption amounts of SFM05905 with increased Fe content were significantly improved compared to SFM118. Both Langmuir and Freundlich models [25,26] were employed to describe the adsorption isotherms. The Freundlich Equation (5) is represented as
(5)$$q_{e}=k_{F}{C_{e}}^{1/n}$$
where q_e_ is the amount of arsenic adsorbed onto the solid phase (mg/g), $C_{e}$ is the equilibrium arsenic concentration in the solution phase (mg/L), $k_{F}$ is roughly an indicator of the adsorption capacity (mg/g), and n is the heterogeneity factor, which has a lower value for more heterogeneous surfaces.

The theoretical Langmuir equation initially developed for gas adsorption on solid surfaces [27] is based on the following assumptions: (1) a fixed number of accessible sites with identical energy levels exist on the adsorbent surface; (2) adsorption is reversible; (3) once an adsorbate occupies a site, no further adsorption can occur on that site; and (4) there is no interaction between adsorbate species. Equation (6) presents the nonlinear form of the Langmuir model:(6)$$q_{e}=\frac{q_{m}k_{L}C_{e}}{\left(1+k_{L}C_{e}\right)}$$
where $q_{e}$ and C_e_ are as previously denoted, k_L_ is the equilibrium adsorption constant related to the affinity of binding sites (L/mg), and $q_{m}$ is the maximum amount of the arsenic per unit weight of adsorbent for complete monolayer coverage.

The Langmuir model is further linearized into four forms as follows [28].

Hanes Woolf linearization:(7)$$\frac{C_{e}}{q_{e}}=(\frac{1}{q_{\max}})\cdot C_{e}+\frac{1}{q_{\max}K_{L}}$$

Lineweaver Burk linearization:(8)$$\frac{1}{q_{e}}=(\frac{1}{q_{\max}K_{L}})\cdot \frac{1}{C_{e}}+\frac{1}{q_{\max}}$$

Eadie–Hofstee linearization:(9)$$q_{e}=(-\frac{1}{K_{L}})\cdot \frac{q_{e}}{C_{e}}+q_{\max}$$

Scatchard linearization:(10)$$\frac{q_{e}}{C_{e}}=-K_{L}\cdot q_{e}+q_{\max}\cdot K_{L}$$
where q_max_ (mg/g) is the maximum saturated monolayer adsorption capacity of an adsorbent, $C_{e}$ (mg/L) is the adsorbate concentration at equilibrium, $q_{e}$ (mg/g) is the amount of adsorbate uptake at equilibrium, and K_L_ (L/mg) is a constant related to the affinity between an adsorbent and adsorbate. For a good adsorbent, a high theoretical adsorption capacity qmax and a steep initial sorption isotherm slope (i.e., high K_L_) are generally desirable [29,30].

The limitations of using these linear forms of the Langmuir model have been outlined by [31] indicating potential errors in parameter estimation due to transformations and assumptions.

The adsorption process is generally studied through an adsorption isotherm curve. The distribution of molecules between liquid and solid phase at the equilibrium state, which is considered as a fundamental factor in determining the sorption capacity [32]. As shown in Table 4, the Freundlich model experimental data are reasonably satisfactory on As (V). The value of 1/n in the Freundlich equation is 0.5 or less for arsenic, which shows high adsorption behavior with SFM05905. In Figure 6, it can be seen that 1/n shows the tendency of adsorption amount, which is different from high affinity and low affinity. When 1/n < 0.5, it indicates that the adsorption amount is similar over a wide concentration range. If it is investigated with a higher concentration, then it can be predicted that As (V) does not increase the maximum adsorption when it changes the initial high concentration, but As (III) will change the adsorption amount with a higher concentration.

Therefore, the Langmuir model’s capability to describe arsenic adsorption by SFM05905 with correlation coefficient (R^2^) is acknowledged, though its limitations in describing adsorption behavior for amorphous materials and assumptions of adsorption occurring on a homogeneous surface are noted [33]. Additionally, the Langmuir isotherm’s capacity for pattern recognition is discussed, with Equation (7) being scrutinized for its fit to modified Langmuir isotherms
(11)$$\frac{C_{e}}{W_{g}}=(\frac{1}{\alpha q_{m}})+(\frac{1}{q_{m}})C_{e}$$
where C_e_ is the equilibrium adsorption concentration (mg/L), W_g_ is the adsorption amount per 1 g of SFM05905 (mg/g), and q_m_ is the maximum amount of the arsenic per unit weight of adsorbent for complete monolayer coverage (mg/g).

In Table 4, the Langmuir adsorption capacities for As (III) range from approximately 30 to 32 mg/g, and for As (V), they range from around 21 to 24 mg/g, respectively, with correlation coefficients. However, when examining the maximum adsorption capacities from the Langmuir model, for As (III), they follow the following order: at 25 °C (31.948 mg/g) > at 45 °C (30.377 mg/g) > at 35 °C (29.815 mg/g), all at 100 rpm. For As (V), the order is as follows: at 25 °C (21.673 mg/g) > at 35 °C (21.711 mg/g) > at 45 °C (24.451 mg/g), all at 100 rpm. This discrepancy may arise from the adsorption process nearing equilibrium at different temperatures, especially at low concentrations. Recent observations question the conclusions regarding the effects of agitation rate on adsorption capacity. For instance, [34] found that the adsorption capacity of vitamin E onto silica varied with the agitation rate, while [35] remarked that agitation rate only affected the speed at which equilibrium was reached, not the equilibrium itself.

In Figure 7, increasing the temperature for As (III) decreased the saturation value (q_m_), while for As (V), it remained unaffected except for a slight change at 45 °C. Thermodynamic parameters (∆G°, ∆H°, ∆S°) of SFM05905 were investigated at temperatures ranging from 25 to 45 °C for As (III) and As (V). Due to the time-dependence of the data and comparison between As (III) and As (V), a kinetic analysis of thermodynamic parameters was infeasible. Consequently, only experimental data could be compared with adsorption isotherms, Freundlich constant (K_F_), and Langmuir constant (K_L_). Thus, we conclude that temperature had no effect on SFM05905 adsorbent for both As (III) and As (V) removal.

## 6. Adsorption Selectivity to As (V) and As (III) for SFM05905 Sample

The distribution coefficient (k_d_) of SFM05905 with respect to anion selectivity in the single element solution is shown in Table 5. The K_d_ value of As (III), As (V), and PO_4_ was particularly high, but the K_d_ value of As (III) and PO_4_ was similar and higher than that of As (V). Since PO_4_ has a high K_d_ value and its chemical behavior is similar to that of As (V), PO_4_ is considered to inhibit As (V) adsorption. In addition, the K_d_ value for SO_4_, F, NO_3_, and CO_3_ was shown to be lower than that of arsenic. It was judged that SO_4_, F, NO_3_, and CO_3_ do not inhibit arsenic adsorption. 

In Table 6, we can compare the variation of K_d_ value for As (III) and As (V) in the binary or ternary elements conditions. The K_d_ value for As (III) and As (V) in the binary condition of the [As (III) + PO_4_] and [As (V) + PO_4_] samples were higher than that of PO_4_, and the K_d_ value for As (III) was higher than that of As (V) under the same conditions of the [As (III) + As (V)] sample.

Simultaneously, in the ternary condition, the k_d_ value of PO_4_ was lower than As (III) and As (V). From the above condition, it was confirmed that the anion selectivity of PO_4_ differs greatly effect on single and mixed conditions of the SFM05905. 

## 7. Adsorption Capacity of As (III), As (V), and PO_4_ by the Column Test

The investigation of arsenic (As) adsorption capacity through a column method is presented. In the context of the ternary adsorbate system illustrated in Figure 8, the adsorption behavior of PO_4_ transitions from adsorption to desorption after 80 h. Accordingly, the adsorption amounts for each component were computed and compared up to the 80 h breakthrough curve.

In the single adsorbate system, the uptake amounts for As (III), As (V), and PO_4_ were determined as 0.032 mmol/g, 0.016 mmol/g, and 0.014 mmol/g, respectively. Meanwhile, in the binary adsorbate system, the uptake amounts for [As (III) + PO_4_], [As (V) + PO_4_], and [As (III) + As (V)] were (0.025 + 0.017) mmol/g, (0.023 + 0.015) mmol/g, and (0.026 + 0.023) mmol/g, respectively. Notably, uptake amounts were influenced by initial concentrations, posing challenges for consistency in single adsorbate systems. Comparisons with other adsorbate systems were also pursued.

Observed trends included near-equivalent PO_4_ uptake amounts in binary systems compared to the single adsorbate system. Conversely, the As (V) uptake increased in the presence of other adsorbates, while the As (III) uptake decreased due to the presence of As (V). In the ternary adsorbate system, the uptake amount of [As (III) + As (V) + PO_4_] was (0.026 + 0.016 + 0.010) mmol/g. As a result, the As (V) uptake remained unchanged, whereas As (III) and PO_4_ uptake decreased relative to the single system. The PO_4_ breakthrough curve displayed consistent shapes across graphs, yet its concentration exceeded the initial concentration in the ternary adsorbate system. This indicated a shift in PO_4_ behavior from adsorption to desorption, contrasting with continuous As (III) and As (V) adsorption.

An analysis of distribution coefficient (K_d_) values revealed distinctions in selectivity, with SFM05905 exhibiting high K_d_ values for PO_4_ and As (III), while As (V) displayed a higher selectivity than PO_4_. The study emphasized that column tests are essential to understanding realistic selectivity, particularly in groundwater or hot spring waters with complex ion compositions. Ultimately, interactions between As (III) and PO_4_ were evident, with As (V) displaying superior selectivity in K_d_ values and remaining unaffected by PO_4_ under mixing condition.

## 8. Assessment of Arsenic Sorption Capacities across Various Adsorbents

As indicated in Table 7, various researchers have explored As (V) adsorption by different adsorbents. In comparing the arsenic adsorption capacities (mg/g) between the Si-Fe-Mg mixed hydroxide containing an SFM05905 granular sample and other reported adsorbents, our study encompasses batch and column methods, examining a range of concentrations from low to high, including standard drinking water (0.01 mg/L) and industrial wastewater (0.1 mg/L). Notably, SFM05905 granular material exhibited superior performance compared to conventional materials like hematite, as well as granular ferric hydroxide (GFH). This suggests that SFM05905 granular material holds promise for arsenic removal in various industrial and drinking water applications. It is important to note that the arsenic adsorption ability of SFM05905 granular material is influenced by its molar ratio. Modifying the molar ratio to decrease Si and Mg and increase Fe could potentially enhance the adsorption process. As a result, efforts have been made to adjust the molar ratio of SFM05905 to optimize its arsenic adsorption capabilities.

## 9. Conclusions

In conclusion, the present study investigated the effectiveness of SFM05905 adsorbent for arsenic removal from aqueous solutions of varying concentrations. The results showed that the adsorption capacity almost reached As (III) 30 mg/g and As (V) 25 mg/g, and the uptake capacity of both was enhanced from low to high concentrations, regardless of temperature. However, the initial concentration was found to be a significant modifier for adsorption capacity. The Langmuir model revealed saturation values of [As (III)-31.948 mg/g] and [As (V) 24.451 mg/g], respectively, with the highest correlation coefficient [As (V), (R^2^ = 0.998), As (III) (R^2^ = 0.981)]. The Freundlich model showed the tendency of the adsorption amount, which differed between high and low affinity. The adsorption anion selectivity of SFM05905 showed low selectivity towards SO_4_, F, NO_3_, and CO_3_ ions, and hence, it was judged that they would not inhibit arsenic adsorption. However, PO_4_ ion was found to inhibit the adsorption of arsenic because of its high selectivity. The adsorption efficiency of SFM05905 was found to be affected by PO_4_ at the batch test, but not in the column adsorption test. The distribution coefficient (K_d_) value was found to be an important factor in groundwater or hot spa water with numerous components or anions. However, it was useful in the column adsorption process. Overall, SFM05905 was found to be an effective adsorbent for the removal of arsenic from wastewater and should be developed as a new material for the removal of pollutants from the environment.

## Figures and Tables

**Figure 1 toxics-12-00280-f001:**
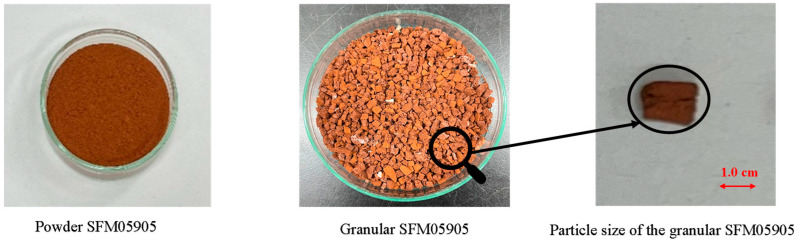
Photograph depicting powder SFM05905, granular SFM05905, and its expansion particles.

**Figure 2 toxics-12-00280-f002:**
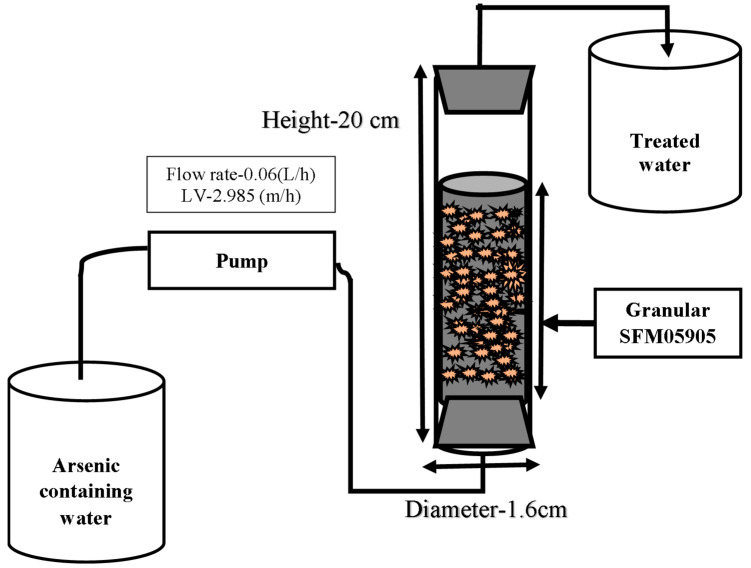
Schematic diagram of the granular SAM05905 adsorption column.

**Figure 3 toxics-12-00280-f003:**
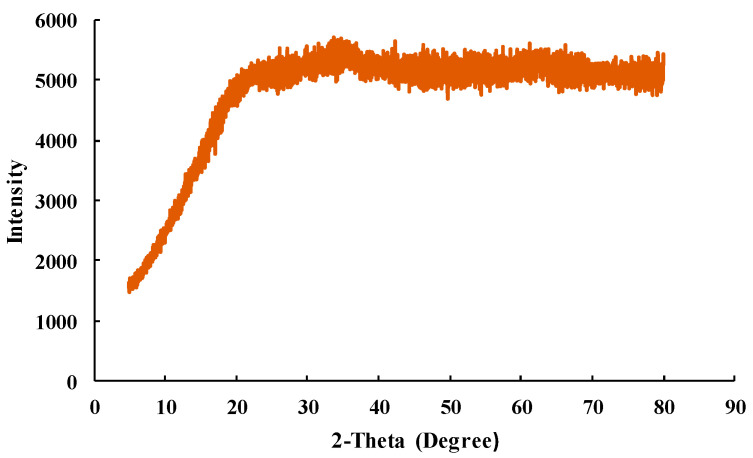
XRD pattern of SFM05905.

**Figure 4 toxics-12-00280-f004:**
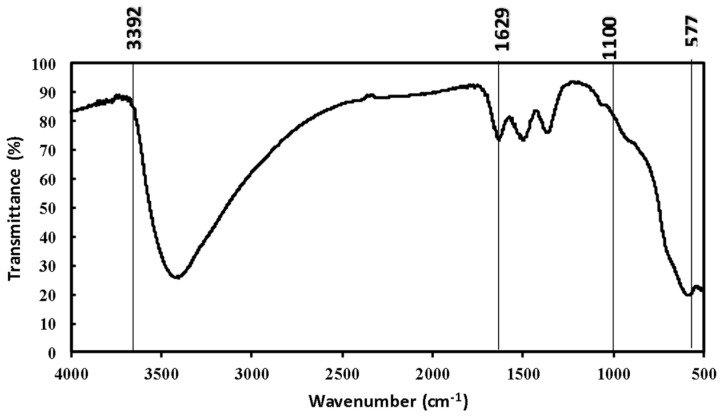
FTIR pattern of SFM05905.

**Figure 5 toxics-12-00280-f005:**
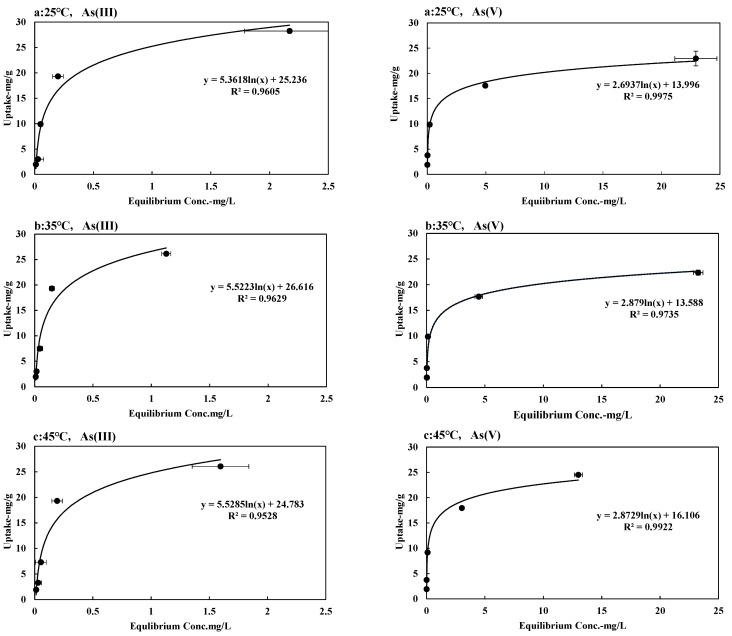
Changes in concentration (10–150 mg/L) of adsorption amount of As (III) and As (V) at several temperatures: a: 25 °C; b: 35 °C; c: 45 °C. ※ Error bars show the Standard deviation.

**Figure 6 toxics-12-00280-f006:**
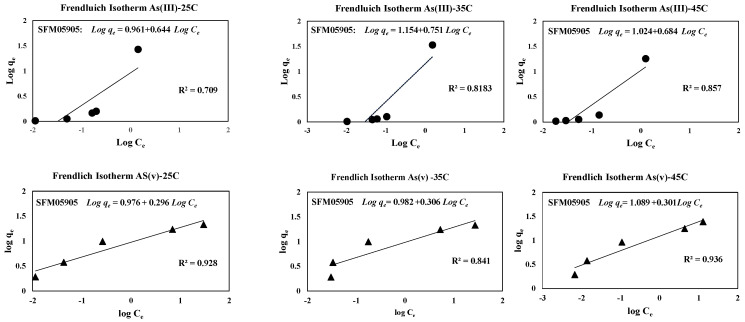
Freundlich isotherm with different conditions of SFM05905 on As (III) and As (V).

**Figure 7 toxics-12-00280-f007:**
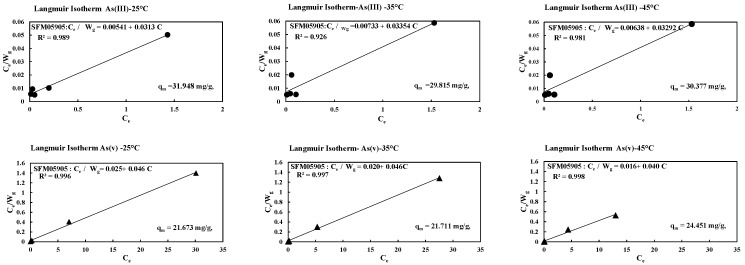
Langmuir isotherm with different temperatures of SFM05905 on As (III) and As (V).

**Figure 8 toxics-12-00280-f008:**
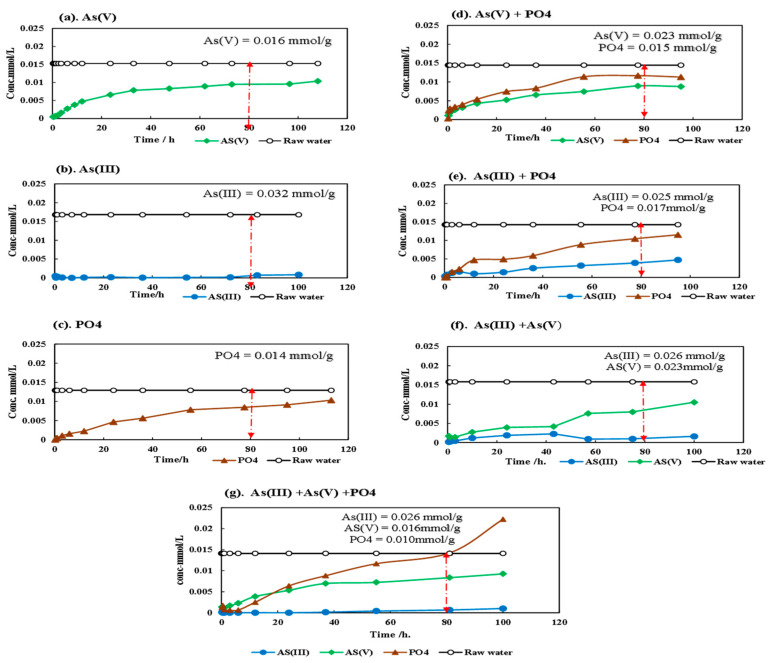
Breakthrough curve of column test with granular SFM05905.

**Table 1 toxics-12-00280-t001:** Used reagents for anion test solutions.

Anion Species	Used Reagent
AsO_3_	Arsenic Standard Solution (As: 1000 mg/L) *
AsO_4_	Na_2_HAsO_4_•7H_2_O
PO_4_	KH_2_PO_4_
SO_4_	Na_2_SO_4_
F	NaF
NO_3_	NaNO_3_
CO_3_	NaHCO_3_

* Commercially available standard solution (Fujifilm Wako Pure Chemical Corp., Osaka, Japan).

**Table 2 toxics-12-00280-t002:** Experimental conditions for column test.

Sample in Aqueous Water (0.015 mmol/L)	Space Velocity (/h)	Packed Granule SFM05905 (g)	Volume of Packed Granular SFM05905 (cm^3^)	Length of Packed Granular SFM05905 (cm)	
				Dry Condition	Wet Condition
As(V)	3.5	2.505	1.03	7.5	8.5
As (III)	3.2	2.516	1.04	8.2	9.4
PO_4_	3.6	2.514	1.04	7.2	8.3
[As (V) + PO_4_]	3.7	2.503	1.03	7.5	8
[As (III) + PO_4_]	3.3	2.515	1.04	8.1	9
[As (III) + As (V)]	3.7	2.513	1.04	7.5	8
[As (III) + As (V) + PO_4_]	3.7	2.504	1.03	7.5	8

**Table 3 toxics-12-00280-t003:** The metal element composition ratio of the powder SFM05905 and the carrying ratio of SFM05905 in its granular form.

		Si	Fe	Mg	Specific Gravity (g/cm^3^)	Carrying Ratio of SFM05905 (%)	Particle Size
Powder SFM05905	Content (mmol/g)	0.50	10.5	0.50	3.8	-	<250 μm
	Obtained composition ratio	0.048	0.907	0.046			
	Targetcomposition ratio	0.05	0.90	0.05			
Granular SFM05905	Content (mmol/g)	Not measured	5.9	Not measured	2.4	56.0	<1 cm^3^

**Table 4 toxics-12-00280-t004:** Arsenic adsorption parameter through Freundlich and Langmuir isotherm and their respective parameters.

As Species	Temperature	Freundlich Model			Langmuir Model		
		K_F_ (mg/g)	1/n	R^2^	K_L_ (L/mg)	q_m_ (mg/g)	R^2^
As (III)	25 °C	9.157	0.644	0.709	5.782	31.948	0.989
	35 °C	14.277	0.751	0.818	4.575	29.815	0.926
	45 °C	10.585	0.684	0.857	5.160	30.377	0.981
As (V)	25 °C	9.477	0.296	0.928	1.834	21.673	0.996
	35 °C	9.609	0.306	0.841	2.218	21.711	0.997
	45 °C	12.284	0.301	0.936	2.539	24.451	0.998

**Table 5 toxics-12-00280-t005:** Distribution coefficient (K_d_) of SFM05905 for various anions in the single element.

K_d_	SFM05905
As (III)	539,000
As (V)	158,000
PO_4_	531,000
SO_4_	300
F	400
NO_3_	200
CO_3_	<1

**Table 6 toxics-12-00280-t006:** Distribution coefficient (K_d_) of SFM05905 for As (III), As (V), and PO_4_ in the binary or ternary element solution.

K_d_	SFM05905			
	As (III) + PO_4_	As (V) + PO_4_	AS (III) + AS (V)	AS (III) + AS (V) + PO_4_
As(III)	122,000	-	42,700	122,000
As(V)	-	36,600	31,200	16,800
PO_4_	7040	8870	-	5030

**Table 7 toxics-12-00280-t007:** Comparative analysis of arsenic sorption capacities among various adsorbents.

No	Adsorbent	Uptake Capacity (mg/g)	Ref.
1	Granular ferric hydroxide (GFH)	2.7	[36]
2	ZnO-GO nanocomposite	8.17	[37]
3	Fe_3_O_4_ nanoparticles	79.49	[38]
4	Hematite	0.381	[39]
5	Metal–organic frameworks (MOFs)	24.83	[40]
6	Fe−BTC MOF	12.287	[41]
7	SUM-8	152.52	[42]
8	Nano-sized TiO_2_	2.58	[43]
9	Biochar Fe-TB	0.91	[44]
10	SFM05905	2.5–25.0	Present study

## Data Availability

The data presented in this study are available on request from the corresponding author.

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
