# Peer review of "Influence of Phosphate on Arsenic Adsorption Behavior of Si-Fe-Mg Mixed Hydrous Oxide"

_toxics, 2024, doi:10.3390/toxics12040280_

Round 1
Reviewer 1 Report
Comments and Suggestions for Authors
The reviewed article 'Influence of Phosphate on Arsenic Adsorption Behaviour of Si- 2 Fe-Mg Mixed Hydrous Oxide' is of interest because of the relevance of the problem of arsenic removal from water . The authors developed a new sorbent for arsenic removal and studied the adsorption process of As(V) and As(III). It has been correctly characterised. However, the authors of the article should supplement/improve certain elements of it. My suggestions are as follows:
1. Amend the first sentence in the 'Introduction' section
2. State the forms in which arsenic is present in solution at the pH under study
3. Table 2. correct the unit for "Space velocity"
4. Consider whether Figure 1 actually shows a "Flow chart..."
5. Currently, it appears that the linear method is no longer suitable for calculating the parameters of the models used (Langmuir). A nonlinear method should be used to determine the parameters of all models used. I suggest reading the article http://dx.doi.org/10.1016/j.watres.2017.04.014, which presents in a simple way how to calculate using such simple and generally available IT tools as e.g. Excel. In many cases, there are large differences when determining constants from linear and non-linear forms.
6. in connection with Note 5, in Figures 5 and 6 it would be preferable to represent isotherms in non-linear form?
7. The literature is quite old. More than half of the literature reports are more than 10 years old.
8. It is suggested to insert a table comparing arsenic removal on different sorbents.
9. the literature record should be corrected
Author Response
Thank you so much for taking the time to review this manuscript . Please find the detailed response below and the corresponding revisions /corrections highlighted /in track changes in the file .

Reviewer 2 Report
Comments and Suggestions for Authors
The manuscript titled ‘Influence of Phosphate on Arsenic Adsorption Behavior of Si-Fe-Mg Mixed Hydrous Oxide’ is a particularly interesting original research article investigating the arsenic adsorption performance of a mixed hydrous oxide containing silicon, iron, and magnesium was investigated, revealing the high anion selectivity for arsenite, arsenate, and phosphate. One of the most interesting issues is the judgement that SO4, F, NO3 and CO3 do not inhibit arsenic absorption of arsenic species. Therefore I would like to strongly recommend the paper for publication after implementing minor corrections. General comments are presented below:
· Silica (Si) is described as a metal several times in the bulk text (eg. Line 209, 220 and more). Silica is considered a semi-metallic element. Consider changing the description of the element while describing several species or other changes clarifying the issue.
Several specific minor comments and questions that should also be considered are listed below:
In Materials and Methods:
· In lines 185-188 please describe the melting method with nitric acid and sulfuric acid.
· Since subchapters 3 and 4 describe the experimental conditions I would suggest changing the numeration to parts 2.4 and 2.5 consequently. Therefore consecutive chapters should be renumbered.
In Tables:
· In Table 3 consider presenting concentration values with uncertainties, preferably with a proper amount of significant digits.
In Results and Discussion:
· In line 284 consider presenting the uncertainties for the estimation of adsorption capacities.
I hope that the Authors will consider minor comments and clarify some issues since the paper might be valuable for a wide scientific audience. That is why I am strongly recommending the paper for publication in Toxics after implementing the above comment.
Author Response
Thank you so much for taking the time to review this manuscript . Please find the detailed response below and the corresponding revisions/corrections highlighted in track changes in the resubmitted files.

Reviewer 3 Report
Comments and Suggestions for Authors
The manuscript toxics-2915406 examines the impact of phosphate on the arsenic adsorption behavior of SiFeMg mixed hydrous oxide (SFM05905). The manuscript falls short on several fronts, from inadequately characterizing the synthesized material to conducting insufficient adsorption studies. Consequently, I do not recommend its publication. Below, I outline the points that support my decision:
1) - In the abstract, the authors use the molar ratio equal to “Si:Fe: Mg = 0.05: 0.9: 0.05 (SFM05905)” and in the introduction used the molar ratio of “Si, Fe and Mg of 0.5:9:0.5”. Suggestion: standardize the way this information is presented. Still, it's unclear how the proportion was chosen. Were there any other variations explored by the authors?
2) Line 114 - “sample pH (6.8) was adjusted by using 1 mol/kg HCl solution and 1 mol/kg NaOH solution…” Why was pH = 6.8 used?
3) Line 118 = “The sample solution was filtered using a membrane filter (HP045AN, Advantec, Tokyo, Japan) with a pore size of 0.45 μm and the arsenic concentration”….Line 141 “The filtrate was collected through a 0.20 μm membrane filter before stirring and after arriving at equilibrium adsorption.” Was the possible adsorption of arsenic on the membranes evaluated? Could this interfere with the process of removing this element?
4) Figure 1. Flow chart of polyacrylamide containing granular SFM05905 and its pore.
The pores in the material aren't visible. Could you please supply a more appropriate image?
5) Line 225 “SFM05905 was analyzed by XRD to assess the physicochemical characteristics of the materials.”
X-ray diffraction analysis is a technique that provides information about the crystallographic structure rather than physicochemical characteristics. Additionally, what contribution does the diffractogram in Figure 3 offer to the study? It appears that this diffraction doesn't provide relevant information, prompting questions about the appropriateness of the analysis or the authors' understanding of the technique being employed. If it's impossible to use XRD for this type of material, there are other structural characterization techniques (such as FTIR, Raman spectroscopy, and others) that could aid in the material's characterization. I also want to emphasize that the material presented (SFM05905) was poorly characterized.
6) The adsorption isothermal study features very few experimental points, a fact evident upon observing Figure 4. Authors often resort to this practice in pursuit of a high R2 value.
To ensure a reliable adsorption study, a greater number of experimental points and various statistical tools, such as the adjusted coefficient of determination, standard deviation, BIC criterion, among others, are utilized. Additionally, it is apparent that the systems have not reached equilibrium, indicating inadequate planning of the study. These characteristics greatly compromise the study presented.
Minor points:
- Line 39 – “. but in anaerobic conditions,”
- Line 42 – Change “Unites state” to “Unites States”
- Line 45 “Inorganic arsenic is more toxic than organic arsenic.” . Please include references to support this statement.
- Line 67 - In the previous study, absorption ability of some SFM samples 68 with different Fe / Mg…”Absorption or adsorption?”
- Line 248 – “KF is roughly an indicator of 248 the adsorption capacity”
- Provide KF unit [(mg g−1(mg L−1)−1/n)]
Comments on the Quality of English LanguageThere are some typos that should be corrected. These were highlighted in the previous item
Author Response
Thank you so much for taking the time to review this manuscript . Please find the detailed responses below attachment file and the corresponding revisions/corrections highlighted / in track changes in the re submitted files.

Round 2
Reviewer 1 Report
Comments and Suggestions for Authors
No comments
Author Response
Please see the attached file of the reviewer comments reply.
